

# Physical health risks of middle-aged people with low social independence: fatal diseases in men, and little attendance to cancer screenings in both sexes

Haruaki Naito[1,2,*], Katsuya Nitta[1,*], Misooja Lee[1], Takeshi Ushigusa[1], Motoki Osawa[2], Takahiro Tabuchi[3] and Yasuhiro Kakiuchi[1,2]

[1] Department of Forensic Medicine, Kindai University, Osakasayama, Osaka, Japan
[2] Department of Forensic Medicine, Tokai University, Isehara, Kanagawa, Japan
[3] Department of Cancer Epidemiology, Osaka International Cancer Institute, Osaka, Osaka, Japan
[*] These authors contributed equally to this work.

Corresponding author
Yasuhiro Kakiuchi,
kakiuchi@med.kindai.ac.jp

## ABSTRACT

**Background**. The existence of social withdrawal (Hikikomori), which meets the conditions "not attending school", "not working", and "isolated at home for more than 6 months", is gradually being discovered by the world, and their mental health and healing is being highlighted. However, there are very few Hikikomori-related surveys searching their physical health, as it is generally believed that most Hikikomori are adolescents. Middle-aged Hikikomori are also found outside Japan, and their physical health is more important, because Hikikomori have difficulty managing their health due to the socially isolated circumstances and lack of sociability. Although "isolated at home for more than 6 months" could not be used, we extracted a group with low social independence with reference to Hikikomori-related surveys. We estimate that people with low social independence have similar characteristics and problems to Hikikomori, because they share many causes for the problem of difficulty in managing their own health. People with low social independence were identified, and their physical health, such as smoking and drinking status, consultation rates of various diseases, and how often they attend cancer screenings, was analyzed.
**Methods**. We extracted middle-aged people with low social independence and a control group from the national survey in Japan and stratified them by sex and age. Their health risks were assessed by univariate analysis. Criteria for the experimental group were set with reference to Hikikomori-related surveys. Criteria for the control group included "aged 40–69", "living with parents", "not receiving care for disabilities", and "working".
**Results**. Low-social-independent men had higher consultation rates for diabetes, stroke or cerebral hemorrhage, myocardial infarction or angina, gastric and duodenum diseases, kidney disease, anemia, and depression, while lower consultation rates for dyslipidemia and hypertension. The tendency of non-smoking and non-drinking was found among them. They seldom attended cancer screenings. Low-social-independent women had higher consultation rates for liver and gallbladder diseases, other digestive

diseases, kidney diseases, anemia, osteoporosis, and depression. The tendency of non-drinking was the same as men. More heavy smokers were found among those aged 40–49 years, with no significant differences in other age groups. They seldom attended cancer screenings, as well as men.

**Conclusions**. In terms of current physical health, low-social-independent men have more fatal diseases. Both sexes with low social independence seldom attend cancer screenings and have an increased risk of developing progressive cancer in the future. At least in terms of non-smoking and non-drinking, they live healthier lives than the control group, and what makes low-social-independent men have various fatal diseases is still unclear.

## BACKGROUND

Hikikomori, a new form of social withdrawal, has been defined as "not attending school", "not working", and "isolated at home for more than 6 months" as described in the guidelines for Hikikomori assessment published in Japan (*Ministry of Health, Labour and Welfare in Japan, 2010*). The existence of Hikikomori has recently been observed not only in Asian countries but also in the U.S., France, Italy, Spain, and Australia (*Kato et al., 2012*; *Teo et al., 2015*; *Furuhashi et al., 2012*; *Wu et al., 2020*; *Malagón-Amor et al., 2018*; *Martinotti et al., 2021*). While the mental health, healing, and prognosis of Hikikomori have been presented as case reports, there are very few studies on their physical health (*Teo & Gaw, 2010*; *Teo, 2013*; *Roza et al., 2020*; *Rooksby, Furuhashi & McLeod, 2020*; *Kubo, Aida & Kato, 2021*).

The reason so little attention is paid to the physical health of Hikikomori is the widespread assumption that most Hikikomori are adolescents and that problems will resolve when they become adults and must work. While this assumption might be true in some countries, it is not the case in others, such as Japan, South Korea, Italy, Spain, and France, where adults are known to live with their parents to some extent. In Japan, an estimated 0.8−1.1% of middle-aged people are Hikikomori living with parents, and many serious social problems occur among them and their elderly parents (*Cabinet Office in Japan, 2019*). The epidemiological distribution of Hikikomori is not known in most countries, and the same situation might happen outside Japan.

Previous surveys have extracted Hikikomori from two perspectives: low sociability and low social independence. In our study, low social independence indicates the socially isolated circumstances without appropriate reasons. Although "isolated at home for more than 6 months", indicating low sociability, could not be used in our study, we consider that people with low social independence have substantially the similar characteristics and problems to Hikikomori. First, as same as Hikikomori, unemployed people without socially appropriate reasons are likely to cause mental stress for the parents, lead to difficulty in asking others for help, and lack money in the household. Second, "isolated at home for

more than 6 months'' is a strict criterion to identify low sociability, and many people in socially isolated circumstances might also show little sociability and be unable to ask others for help, and manage their physical health during the parents' lifetime and after the parents fall ill or die. From the national survey data, we set the criteria with reference to the survey for middle-aged Hikikomori, extracted a group with low social independence, compared their characteristics to those of middle-aged Hikikomori, and analyzed their physical health.

## METHODS

### Data collection

We used data from the Comprehensive Survey of Living Conditions (CSLC) conducted by the Ministry of Health, Labor, and Welfare of Japan on June 6, 2019, in which 535,619 people participated. CSLC is a large-scale national survey every 3 years to see Japanese living condition in terms of health and economy, and the questionnaires are distributed to people chosen by simple random sampling. The characteristics of the respondents were collected by the household questionnaire and health questionnaire and could be linked by matching the household number and individual number on both questionnaires.

### Definition of experimental and control groups in this study

In general, "not attending school", "not working", and "isolated at home for more than 6 months" were adopted as the definition of Hikikomori (*Ministry of Health, Labour and Welfare in Japan, 2010*). In this study, "aged 40–69 years" and "living with parents" were set as inclusion criteria of experimental group. To extract a group with a low degree of social independence, "attending school", "working", "doing household chores substitute for working", "seeking employment", "head of the household", "caring for household members", "married", "having a child", and "receiving care for disabilities" were set as exclusion criteria with reference to the middle-aged Hikikomori-related survey (*Cabinet Office in Japan, 2019*).

As inclusion criteria of control group, "aged 40–69 years", "living with parents", and "working" were set. Exclusion criterion was "receiving care for disabilities". "Aged 40-69 years", "living with parents", and "not receiving care for disabilities" were set as common conditions between the experimental and control groups in terms of health risks and access to medical support. As mentioned above, since the core of the experimental group is the low degree of social independence, "working" was set in the control group for some social independence.

### Explanatory variables

As explanatory variables, we set smoking status; drinking status; current consultation for diabetes, obesity, dyslipidemia, hypertension, depression, stroke or cerebral hemorrhage, myocardial infarction or angina, other cardiovascular diseases, digestive diseases (gastric, duodenal, liver, gallbladder, and other digestive diseases), kidney diseases, anemia, osteoporosis, or cancer; and participation in cancer screening (gastric, lung, colon, breast, and uterus) over 1–2 years. Smoking status was defined as following categories: heavy

smokers (smoking >=20 cigarettes per day), moderate smokers (currently smoking, but not categorized as heavy smokers), and abstainers (never smoking or not smoking over a month). Comparing heavy smokers and abstainers, adjusted hazard ratios for cancer mortality were reported as 2.32 (all sites), 9.86 (lung), and 5.45 (upper aero-digestive tract) (*Meyer et al., 2015*). Drinking status was defined as following categories: heavy drinking (more than four drinks on any day or more than 14 drinks per week for men, and more than three drinks on any day or more than seven drinks per week for women), moderate drinking (currently drinking, but not heavy drinking), abstainers (hardly/never drinking), and unknown (unknown alcohol consumption). The definition of drinking status is based on National Institute on Alcohol Abuse and Alcoholism (*National Institute on Alcohol Abuse and Alcoholism, n.d.*).

## Statistical analysis

We conducted univariate case-control analysis to examine the characteristics of the experimental and control groups using EZR v1.54. *P*-value and Odds ratio are based on 95% confidence interval. Participants who did not respond to even one of the relevant explanatory variables were excluded from the analysis.

## Ethics approval

The ethics committee of Kindai University approved the analysis of human participants data in this study (R03-139). All participants received informed consent verbally before they filled in the questionnaires.

# RESULTS

## Validity of sample

Figure 1 shows the flowchart of the participants selection. We extracted the experimental group to analyze people with low social independence, including Hikikomori. Whether this group has similar characteristics to Hikikomori can be explained by comparing to the middle-aged Hikikomori survey about the distribution and psychiatric health. First, from the survey, an estimated 1.1% and 0.3% of middle-aged people in Japan are middle-aged male Hikikomori and female Hikikomori (*Cabinet Office in Japan, 2019*). In Fig. 1, the prevalence of low-social-independent men among the control group in the CSLC data is 2.0% (796/39,454), while that of low-social-independent women is 0.9% (347/39,454). Comparing to the other studies reporting that the sex ratio of Hikikomori was ranging from about 2:1 to 3:1, this result is consistent with them (*Tajan, Hamasaki & Nancy, 2017*; *Cabinet Office in Japan, 2019*; *Yong & Nomura, 2019*; *Koyama et al., 2010*). Second, odds ratio in depression between middle-aged Hikikomori and the control group is 7.9 in the previous survey, and that in our result is 5.6–13.2 (*Cabinet Office in Japan, 2019*). Although we could not include ''isolated at home for more than 6 months'' as a criterion, our subjects were estimated to have similar degrees of psychological issues to Hikikomori. Since prolonged depression leads to loss of sociability and cognitive function, if people with low social independence have a high prevalence of depression, it is likely to cause problems as same as Hikikomori. Therefore, we estimate that our extraction is valid to a certain

extent for the purpose of analyzing groups with similar characteristics to middle-aged Hikikomori.

### Univariate analysis

Table S1 shows the characteristics of low-social-independent men and control group, examined by univariate analysis. About lifestyle, low-social-independent men smoked less and drank less. About current consultation, they had higher consultation rates for diabetes, stroke or cerebral hemorrhage, myocardial infarction or angina, gastric and duodenum diseases, kidney diseases, anemia, and depression, while lower consultation rates for dyslipidemia and hypertension. They attended all cancer screenings (gastric, lung, and colon) less.

Table S2 shows the characteristics of low-social-independent women and control group. About lifestyle, low-social-independent-women drank less. The prevalence of heavy smokers was significantly high among those aged 40–49, and no significant differences about smoking were observed in other age groups. About current consultation, they had higher consultation rates for liver and gallbladder diseases, digestive diseases (except gastric, duodenal, liver, and gallbladder diseases), kidney diseases, anemia, osteoporosis. They attended all cancer screenings (gastric, lung, colon, breast, and uterine) less, as same as low-social-independent men.

## DISCUSSION

### Common characteristics in men and women

The subjects in this study, "low-social-independent people" equate to the middle-aged people living with parents with some criteria based on Hikikomori-related surveys: "not attending school", "not working", "the reason for not working is not household chores", "not seeking employment", "not head of the household", "not caring for household members", "not married", "not having a child", and "not receiving care for disabilities" (*Cabinet Office in Japan, 2019*).

First, it was unexpected that low-social-independent people lead a healthy lifestyle in terms of non-drinking, although our target subjects are middle-aged. We do not assume that the presence of colleagues at work in the control group is the main reason for this difference in drinking. In many surveys, outing without communication, such as shopping, is not excluded from the definition of Hikikomori, and most of them are known to go to places like convenience stores (*Cabinet Office in Japan, 2019*; *Hamasaki et al., 2022*; *Tateno et al., 2019*). Therefore, even if our subjects were consistent with Hikikomori, they would not have had much trouble going to those places and buying some alcohol and cigarettes. In 2019, although up to 70% of alcohol consumption occurs at home in Japan, low-social-independent people tend not to consume alcohol in this result (*Statistics Bureau in Japan, 2021*). This might indicate ease of access to alcohol is unlikely to affect the drinking tendency among low-social-independent people. The COVID-19 pandemic increased opportunities for drinking at home worldwide (*The Brewers of Europe, 2021*; *Statistics Bureau in Japan, 2022*). The tendency of non-drinking among low-social-independent people in Japan might not change because the tendency is originally regardless of access to alcohol. However, in

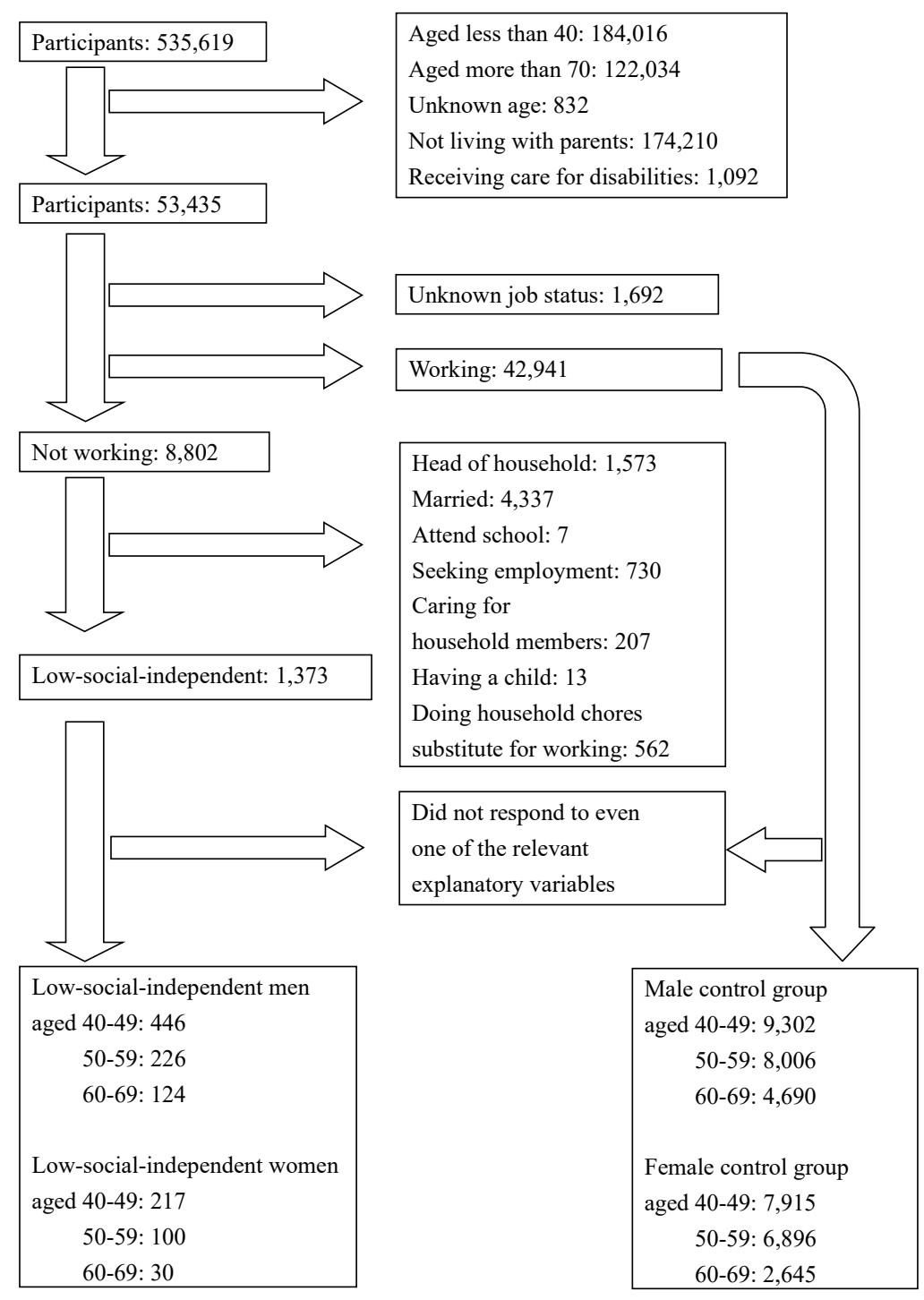

**Figure 1  Flowchart of the participants selection.**

the countries where the difficulty of access to alcohol is the only limitation on their alcohol use, the relaxation of restrictions in the pandemic would have resulted in a corresponding surge in their alcohol consumption.

Smoking prevalence among low-social-independent men was significantly low. No significant smoking prevalence was observed among low-social-independent women except for those aged 40–49 years. However, smoking prevalence among women is low worldwide, particularly in Japan (approximately 7%) (*OECD, 2019*). We consider low-social-independent women have non-drinking tendency, because Table S2 shows the smoking prevalence among those aged 40-69 years was approximately 9%. The price of cigarettes in Japan is among the lowest in the world (*Cahn et al., 2018*). Therefore, easy access to cigarettes is unlikely to affect smoking tendency among them. It is unclear whether the non-drinking and non-smoking tendency is a universal lifestyle for people with low social independence and what contributes to this phenomenon. A recent study shows that the COVID-19 pandemic could cause an increase in the number of Hikikomori (*Rooksby, Furuhashi & McLeod, 2020*). Further observation is necessary to see how the pandemic affects the situation around them over the world.

Second, it is important to note that the low-social-independent people attend cancer screenings much less. 10–35% of them attended at least one cancer screening within 1–2 years, compared with 40–60% of participants in the control group. Especially, when they were under 60 years old, only 10–20% of them attended. Before our study, we had expected them to have a sense of crisis regarding their health and to have some access to medical care, as they live with their elderly parents. However, the results of our study show the opposite. Low-social-independent people do not see doctors in terms of prevention or screening, because they might lack the knowledge about diseases, avoid discussing health problems with their family, or just be overwhelmed by their current psychological issues. However, we doubt whether cancer tend to occur in Hikikomori. They smoke less and drink less, while smoking is attributed to about 20–40% of cancer death (*Katanoda et al., 2021*; *Lortet-Tieulent et al., 2016*). Therefore, the situations about cancer among them might be mainly divided into two, cancer-free or quite progressive.

**Men**

Lower consultation rates for dyslipidemia and hypertension were observed. Low consultation rate does not equate to low prevalence of disease, especially when the study includes participants who rarely see doctors. In this result, the consultation rate of these diseases among working people largely increased at their 50s, while that among non-working people changed little between their 40s and 50s. Therefore, about hypertension and dyslipidemia, the low consultation rate among the experimental group is likely to be explained by the lack of medical checkup at workplace.

As mentioned above, we can assume dyslipidemia and hypertension to have been long-term unknown underlying conditions among low-social-independent men. It also explains that they have higher consultation rates for fatal diseases like diabetes, stroke or cerebral hemorrhage, myocardial infarction or angina, and kidney diseases. In contrast, they tend not to smoke or drink, and the causes of those lifestyle-related diseases are still

unclear. Further, whether the cause of the low social independence contributes to their bad health and poor medical access remains unknown. While withdrawal in youth or an unsuccessful job search led to long-term withdrawal with very low social independence, problems in workplace relationships could lead to short-term withdrawal with some social independence.

## Women

Only depression, anemia and kidney diseases are common among both sexes with low social independence, whereas digestive diseases (except gastric and duodenal diseases) and osteoporosis are prevalent among women. We estimate that lifestyle diseases are not latent in women more than men. First, many fatal diseases related to lifestyle diseases, such as diabetes, stroke or cerebral hemorrhage, and myocardial infarction or angina, were only observed in men. Second, unlike men, the consultation rate of dyslipidemia and hypertension largely increased at their 50s, indicating that women might go to medical checkup more than men under the same socially isolated circumstances. Besides, low-social-independent women had higher consultation rate for almost all cancer screenings than men. Most of "digestive diseases" might indicate irritable bowel syndrome, considering its prevalence in women and its relationship with depression (*Sugaya, Nomura & Shimada, 2012*; *Cho et al., 2011*; *O'Mahony et al., 2009*). The diseases prevalent in low-social-independent women would be less harmful than the fatal diseases prevalent in men, and there are striking sex differences in the physical health of low social independent people. The tendency of non-smoking and non-drinking is the same in both sexes, and other factors might contribute to this result. For example, estrogen would contribute significantly to women's health. Estrogen is known to prevent atherosclerotic development and some related diseases like hypertension, dyslipidemia, ischemic heart diseases and chronic kidney diseases (*Fontaine et al., 2020*; *Farahmand et al., 2021*). However, it is unclear whether only estrogen can explain the striking sex difference. We cannot dismiss the possibility that low-social-independent women have better lifestyles than men. More surveys are necessary to seek what makes low-social-independent men have fatal diseases, focusing on their diet, exercise habit, length and the cause of the low social independence.

This study had some limitations. First, our criteria lack the condition, "isolated at home for more than 6 months", which is a part of the precise definition of Hikikomori. We consider that many people with low social independence would show low sociability and have substantially the same characteristics and problems as Hikikomori, even if some of them do not match the precise definition. However, "isolated at home for more than 6 months" can directly identify people with low sociability. Our sample includes people with some sociability, who are likely to manage their own and their parents' health and ask others for help. The physical health of such people would be less important than people with low sociability. If "isolated at home for more than 6 months" had been set as criterion in this study, lower levels of consultation for diseases and screening for cancer would have been observed. Hence, the low cancer screening rate in the experimental group may be underestimated. Second, some of the diseases surveyed in CSLC data are unclear. We assumed that "digestive diseases" prevalent in low-social-independent women is not

harmful and not related to lifestyle diseases, compared with various fatal diseases in men. However, this is not an accurate discussion. Third, we could not compare our results because of the scarcity of studies on Hikikomori's physical health. Whether the physical health of low-social-independent people including Hikikomori is important depends on the proportion of them in each country. We believe that this study contributes to a basic understanding of low-social-independent people.

## CONCLUSIONS

Interestingly, except anemia, kidney diseases and depression, low-social-independent men and women tend to seek medical attention for entirely different diseases. Men have higher consultation rates for diabetes, myocardial infarction or angina, stroke or cerebral hemorrhage, and kidney diseases, which are fatal and related to lifestyle diseases. Significantly lower consultation rates for dyslipidemia and hypertension were found, but we estimate that these lifestyle diseases were latent in them for long time and led the various fatal diseases. On the other hand, women have higher consultation rates for anemia, osteoporosis, kidney diseases, and digestive diseases (except gastric and duodenal diseases). Hence, low-social-independent men have increased mortality risks in terms of current physical health. As for the future physical health, both sexes seldom attend cancer screenings. Although the prevalence of overall cancer might be low due to their lifestyles of non-drinking and non-smoking, we consider that progressive cancers are frequently latent in them.

## ACKNOWLEDGEMENTS

We would like to thank Honyaku Center Inc. for English language editing.

### Funding
This work was supported by the Ministry of Education, Culture, Sports, Science, and Technology (No. 19K19365), and the Yuumi Memorial Foundation's grant for home health care issue research. The funders had no role in study design, data collection and analysis, decision to publish, or preparation of the manuscript.

### Grant Disclosures
The following grant information was disclosed by the authors:
Ministry of Education, Culture, Sports, Science, and Technology: 19K19365.
Yuumi Memorial Foundation's grant for home health care issue research.

### Competing Interests
The authors declare there are no competing interests.

## Author Contributions

- Haruaki Naito conceived and designed the experiments, analyzed the data, authored or reviewed drafts of the article, and approved the final draft.
- Katsuya Nitta conceived and designed the experiments, analyzed the data, authored or reviewed drafts of the article, and approved the final draft.
- Misooja Lee analyzed the data, prepared figures and/or tables, and approved the final draft.
- Takeshi Ushigusa analyzed the data, prepared figures and/or tables, and approved the final draft.
- Motoki Osawa conceived and designed the experiments, authored or reviewed drafts of the article, and approved the final draft.
- Takahiro Tabuchi conceived and designed the experiments, authored or reviewed drafts of the article, and approved the final draft.
- Yasuhiro Kakiuchi conceived and designed the experiments, analyzed the data, authored or reviewed drafts of the article, and approved the final draft.

## Human Ethics

The following information was supplied relating to ethical approvals (*i.e.*, approving body and any reference numbers):

Kindai University granted Ethical approval to carry out the study by analyzing the national data including human participants data.

## Data Availability

The raw data is available in the Supplementary File.

## Supplemental Information

Supplemental information for this article can be found online at http://dx.doi.org/10.7717/peerj.14904#supplemental-information.

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
