# Peer review of "Physical health risks of middle-aged people with low social independence: fatal diseases in men, and little attendance to cancer screenings in both sexes"

_PeerJ, doi:10.7717/peerj.14904_

## Round 0.1 · original submission · Major Revisions

This manuscript will require significant revision and clarification of the fundamental assumptions underlying the study. While I believe the issues raised are addressable, the revised version will need re-review before a final decision can be issued. As you will note, all three reviewers identified substantial issues with respect to the identification and classification(s) of Hikikomori people in the study. In particular, Reviewer 2 questioned the defining criteria used for Hikikomori, which will need to be resolved before the revised manuscript could be considered for publication. I hope you will give due consideration to the thoughtful and thorough comments of the reviewers, and that you will consider re-submitting a revised version of this work to PeerJ. Please note that if you decide to submit a revision, each item raised by the reviewers will need to be specifically addressed in the new report. Thank you again for submitting your research.

Reviewer 1 ·

Basic reporting

Thank you for allowing me to contribute to the review of the paper entitled “Physical health risks of middle-aged social withdrawal (Hikikomori): Fatal diseases in men, and little attendance to cancer screenings in both sexes” I believe this manuscript focused on an essential but insufficiently known topic and provided valuable data. I will aim to be brief and constructive in my comments.

Background
1. I want the authors to explain what exactly is meant by “sociability,” which is the core of hikikomori in this study. Furthermore, the reasons why you can postulate that the core of hikikomori is the lack of sociability need to be provided.

Experimental design

Methods
1. The authors defined the control group as “aged 40-69 years,” “living with parents,” and “not receiving care for disabilities.” However, I did not understand that only these conditions were sufficient to consider the control group “working.” If the authors had other conditions in the control group, I suggest that the authors accurately describe them in the text. Also, consider dividing the criteria for both groups into inclusion and exclusion criteria.

2. Provide the evidence for the categories classified by drinking status (Hardly, never, or over 1-3 times a month) and smoking status (Abstainer, Moderate smoker, Heavy smoker).

Results
3. Among all participants, the percentage of the hikikomori group seems to be considerably larger among women (739/9748) than among men (926/20686). I wonder how the authors recognize the reason for this tendency, as it is generally known that hikikomori people are more likely to be men (Nonaka et al., 2022). Additionally, given the condition of “do not care for their household or children,” it seems that “housewife” was excluded from the hikikomori group. If so, the hikikomori group should have a more significant percentage of men, but this is not the case in this study.

Nonaka, S., Takeda, T., & Sakai, M. (2022). Who are hikikomori? Demographic and clinical features of hikikomori (prolonged social withdrawal): A systematic review. Australian & New Zealand Journal of Psychiatry, 00048674221085917.

Limitation
4. The authors described the lack of hikikomori conditions for avoiding communication with others and going outside as a limitation of this study. I recommend discussing the influence of the lack of this condition on the results. For example, if avoiding communication with others and going outside were included in the conditions, could it be assumed that the hikikomori individuals would be even less likely to be screened for cancer or to be diagnosed with some diseases?

Validity of the findings

1. The household and health questionnaires seem to be different questionnaires. How were the responses to the household and health questionnaires linked?

Additional comments

Discussion
1. The discussion on the low consumption of alcohol among hikikomori people is fascinating. I suggest the authors further discuss whether the low smoking consumption is due to the same or other reasons.

2. Provide the publication year of the following citations in the text:
(Cabinet Office in Japan)
(Ministry of Health, Labour and Welfare in Japan.)
(Statistics Bureau in Japan)

Reviewer 2 ·

Basic reporting

.

Experimental design

.

Validity of the findings

.

Additional comments

This study is focusing on the hikikomori and physical condition based on population based dataset. The crucial error in this study is the mistake of the definition of hikikomori. "Not going out for more than 6 months" is the mandatory criteria of hikikomori, but in this study, the authors have completely ignored this aspects. Thus, this study is dangerous to be published in public.

·

Basic reporting

no comment

Experimental design

no comment

Validity of the findings

- In the discussion, lines 142-143, I don't quite understand why it is assumed that in other countries alcohol consumption is lower than in Japan.
- In lines 159-162, it is described that hikikomori "are known to seek psychiatric help", which is not usual, but rather the opposite, as described in previous studies. Usually patients with hikikomori do not seek psychiatric help, and it is usually the families who consult for help.

---

## Round 0.2 · Major Revisions

As you will see, although the reviewer noted considerable improvements in the revised manuscript, there are several substantive questions which must be addressed before a final decision can be rendered. I encourage you to carefully consider the reviewer's comments, and to address the points raised in your response. Thank you again for considering PeerJ as a forum for your work.

Reviewer 1 ·

Basic reporting

It is unclear what relationship the authors assumed theoretically between ‘people with low social independence’ and hikikomori. For example, the authors need to indicate in the Background whether they assumed continuity or they assumed what is the same and what is partially different between the two concepts. In addition, why does 'low social independence' not include people who live with their parents? Defining social independence may help answer this question.

Experimental design

no comment

Validity of the findings

This study attempts to provide data on physical health by sex; therefore, it should also evaluate the validity of the sample based on the proportion of sex differences. The authors responded that they disagreed with the findings on the sex ratio in hikikomori because of bias in data collection. However, sex differences have also been reported in studies randomly selected from community populations (Koyama et al., 2010; Tajan et al., 2017), not only in studies that the authors noted were “biased.” Cabinet Office in Japan (2019) showed a similar trend.
Why did this study show a different trend from these consistent sex ratio findings? Did this different trend influence the results of this study?

---

## Round 0.3 · accepted · Accept

Prior to final publication, I recommend changing the sentence in the abstract which reads: "Low-social- independent women had higher consultation rates for liver and gallbladder diseases, digestive diseases (except gastric, duodenal, liver, and gallbladder diseases)..."
to read: "liver and gallbladder diseases, other digestive diseases...", as this is a confusing passage in English. I do not feel that this should delay acceptance, and I look forward to reading the published version of this work. Thank you for choosing to submit your work to PeerJ.

Reviewer 1 ·

Basic reporting

I believe the authors have appropriately responded to my comments and revised the manuscript.

Experimental design

no comment

Validity of the findings

no comment